# Characterization of DAG Binding to TRPC Channels by Target-Dependent *cis–trans* Isomerization of OptoDArG

**DOI:** 10.3390/biom12060799

**Published:** 2022-06-07

**Authors:** Hazel Erkan-Candag, Denis Krivic, Mathias A. F. Gsell, Mina Aleksanyan, Thomas Stockner, Rumiana Dimova, Oleksandra Tiapko, Klaus Groschner

**Affiliations:** 1Gottfried Schatz Research Center for Cell Signaling, Metabolism and Aging—Division of Biophysics, Medical University Graz, 8010 Graz, Austria; hazel.erkan@medunigraz.at (H.E.-C.); denis.krivic@medunigraz.at (D.K.); matthias.gsell@medunigraz.at (M.A.F.G.); 2 Max Planck Institute of Colloids and Interfaces, 14476 Potsdam, Germany; mina.aleksanyan@mpikg.mpg.de (M.A.); rumiana.dimova@mpikg.mpg.de (R.D.); 3Institute for Chemistry and Biochemistry, Freie Universität Berlin, 14195 Berlin, Germany; 4Institute of Pharmacology, Center for Physiology and Pharmacology, Medical University of Vienna, 1090 Vienna, Austria; thomas.stockner@meduniwien.ac.at

**Keywords:** lipid photopharmacology, TRPC channels, diacylglycerols, photoisomerization

## Abstract

Azobenzene-based photochromic lipids are valuable probes for the analysis of ion channel–lipid interactions. Rapid photoisomerization of these molecules enables the analysis of lipid gating kinetics and provides information on lipid sensing. Thermal relaxation of the metastable *cis* conformation to the *trans* conformation of azobenzene photolipids is rather slow in the dark and may be modified by ligand–protein interactions. *Cis* photolipid-induced changes in pure lipid membranes as visualized from the morphological response of giant unilamellar vesicles indicated that thermal *cis–trans* isomerization of both PhoDAG-1 and OptoDArG is essentially slow in the lipid bilayer environment. While the currents activated by *cis* PhoDAG remained stable upon termination of UV light exposure (dark, UV-OFF), *cis* OptoDArG-induced TRPC3/6/7 activity displayed a striking isoform-dependent exponential decay. The deactivation kinetics of *cis* OptoDArG-induced currents in the dark was sensitive to mutations in the L2 lipid coordination site of TRPC channels. We conclude that the binding of *cis* OptoDArG to TRPC channels promotes transition of *cis* OptoDArG to the *trans* conformation. This process is suggested to provide valuable information on DAG–ion channel interactions and may enable highly selective photopharmacological interventions.

## 1. Introduction

Recent advances in lipid photopharmacology enabled new insights into lipid signaling pathways, protein–lipid interactions, and the identification of lipid binding sites in signaling molecules [1,2,3]. This strategy of attaining control over biological functions with exceptional spatiotemporal precision using light-induced isomerization of synthetic photochromes has been developed over the past decade, and its value for biology and medicine is increasingly recognized [4,5,6,7].

Nonetheless, the molecular principles underlying photopharmacology are complex and incompletely understood. In general, the photopharmacological concept is based on introducing unique photochemical features into biologically active small molecules, resulting in the ability to control their conformational state in terms of *cis–trans* isomerization rapidly and with a reasonable switching amplitude in the photostationary state [5,6]. This is most commonly achieved by introducing an azobenzene photochrome into the active ligand structure [8]. Fast photoisomerization triggered by illumination at the appropriate wavelength enables exposing a given biological target almost instantaneously and with outstanding spatial precision to a ligand with defined conformation and conformation-dependent affinity and efficacy (intrinsic activity). Hence, photoisomerization enables spatiotemporal control over affinity and/or efficacy of drug-like molecules by light.

It has been repeatedly demonstrated that both conformational states (*cis* and *trans*) of the photochromes are potentially able to exert biological effects with defined affinity and efficacy [8,9]. For recently developed azobenzene-based photochromic lipids, biologic activity was found primarily linked to the *cis* conformation [2,10], which, for standard azobenzenes, is thermodynamically less stable and returns to the *trans* state via thermal relaxation with variable kinetics. Rapid *trans–cis* photoisomerization of azo-diacylglycerols (DAGs) enables a “concentration clamp” scenario with essentially fast alterations in the active ligand concentration at the target. This “optical lipid clamp” strategy was found suitable to delineate the kinetics of DAG activation of such targets as TRPC channels and explore the structural basis of channel regulation as well as mechanistic aspects of channel regulation by the lipid [2]. Nonetheless, such analysis is complicated by the difficulty to clearly distinguish between “affinity” and “efficacy” switch mechanisms and by an overlap of these principles for many photopharmacological actions as illustrated in Figure 1. Control over signaling proteins by a photochromic ligand is likely to involve isomerization steps of ligands within their binding pockets. It has recently been demonstrated that the protein environment exerts a significant impact on the photoisomerization of bound ligands [11]. Similarly, the molecular environment of photochromic ligands is expected to impact their thermal relaxation behavior. Hence, a detailed characterization of light-induced changes in protein function may provide useful information on ligand-binding pockets in proteins and reveal the structure–function relations in the target protein complexes. Importantly, this approach requires consideration of the photochemical characteristics of the ligands within the surroundings of the target protein. For photoswitchable lipids, little is so far known about their isomerization characteristics within the lipid bilayer (see, e.g., [12,13,14]), even though biomembranes represent the main application platform in which these molecules are used to control transmembrane- or membrane-associated protein targets.

Here, we report that photochromic diacylglycerol OptoDArG displays prominent *cis–trans* relaxation when bound to TRPC channels. The kinetics of thermal relaxation of this photochromic channel activator was found dependent on the channel isoform and the amino acid sequence of the predicted lipid-binding domain. We propose analysis of the kinetics of photolipid-mediated modification of channel functions as a strategy to identify structures and principles underlying their lipid regulation. Moreover, the TRPC isoform-dependent thermal relaxation of channel activator *cis* OptoDArG reported here may serve as the basis of novel protocols for selective photopharmacological control over TRPC conductance.

## 2. Materials and Methods

### 2.1. DNA Constructs and Reagents

All the reagents used were of molecular biology grade, purchased from Merck (Vienna, Austria) unless specified otherwise. The following constructs were generated: the YFP-TRPC3, YFP-TRPC6, and CFP-TRPC7: hTRPC3 (Q13507-3) and hTRPC6 (Q9Y210-1) genes were cloned into the peYFP-C1 and peCFP-C1 vectors using EcoRI/XbaI cloning sites; hTRPC3 peYFP-C1 was used as a template for generating TRPC3 mutants. The TRPC7–mseCFP construct (Q9WVC5-1 cloned into the pCI-neo vector) was kindly provided by Prof. Yasuo Mori (Kyoto University, Japan); PhoDAG-1 (CAS No. 1985595-31-0) was from Avanti Polar Lipids (Birmingham, USA). OptoDArG (CAS No. 2230617-93-1) was from Aobious (Gloucester, MA, USA). For GUV experiments, 1-palmitoyl-2-oleoyl-glycero-3-phosphocholine (POPC; CAS No. 26853-31-6) was procured from Avanti and LiChrosolv^®^ chloroform (CAS No. 67-66-3; Avanti Polar Lipids, Birmingham, USA) was procured from Merck (Vienna, Austria, while NaCl (CAS No. 7647-14-5), sucrose (CAS No. 57-50-1), and glucose (CAS No. 50-99-7) were procured from Merck (Vienna, Austria).

### 2.2. Cell Culture and Transfection

Human embryonic kidney 293 (HEK293) (CLS 300192; CLS, Koeln, Germany) cells were cultured in Dulbecco’s modified Eagle’s medium (DMEM, D6429, Invitrogen, Thermo Fisher Scientific; Massachusetts, USA) with 10% supplementation of fetal bovine serum (FBS), HEPES (10 mM/L), L-glutamine (2 mM/L), streptomycin (100 μg/mL), and penicillin (100 U/mL) at 37 °C and 5% CO_2_. The cells were authenticated by STR and regular tests were performed to confirm the lack of contamination with mycoplasma. For transfection, the media was aspirated and the HEK293 cells were rinsed with PBS. The cells were incubated with Accutase (250–500 μL) for 5 min at 37 °C. Detached cell suspension was mixed with fresh DMEM in two times the volume of Accutase; 1 × 10^5^ cells suspension was centrifuged at 300× *g* for 2 min. The supernatant was discarded, and the cell pellet was suspended in serum-free medium (60 μL). The cells were transiently transfected with 1 μg plasmid DNA using PolyJet (SignaGen Laboratories; Frederick, USA) according to the manufacturer’s protocol. The cells were seeded on 6 × 6 mm glass coverslips and the medium was changed after 6 h incubation.

### 2.3. Electrophysiology

The transfected HEK293 cells were seeded on glass coverslips the day before the experiments. After 24 h, the coverslips were mounted in a perfusion chamber on an inverted microscope (Zeiss Axiovert 200M; Munich, Germany) with a 40×/0.75 objective. CoolLED pE-300Ultra (CoolLED; Andover, England) was used as an excitation source. The transfected cells were detected by illumination at 490 nm wavelength. Patch clamp recordings were performed in whole-cell configuration using an Axopatch 200B amplifier (Molecular Devices, San Jose, CA, USA) connected with a Digidata-1550B Digitizer (Axon Instruments; Molecular Devices, San Jose, CA, USA). The signals were low-pass filtered at 2 kHz and digitized with 8 kHz. For photopharmacological measurements, the coverslips were transferred to a perfusion chamber filled with 20 µM OptoDArG (Aobious; Gloucester, MA, USA); then, the cells were illuminated with UV (365 nm) or blue (430 nm) light or kept in the dark depending on the protocol. The voltage clamp extracellular solutions contained: 140 mM NaCl, 10 mM HEPES, 10 mM glucose, 2 mM MgCl_2_, 2 mM CaCl_2_; pH was adjusted to 7.4 with NaOH. The pipette solution contained: 150 mM cesium methanesulfonate, 20 mM CsCl, 15 mM HEPES, 5 mM MgCl_2_, 3 mM EGTA; titrated to pH 7.3 with CsOH. Thin-wall capillary pipettes made with borosilicate glass with filament (Harvard Apparatus; Massachusetts, USA) were pulled to a resistance of 3–4 MΩ.

### 2.4. Giant Unilamellar Vesicle (GUV) Preparation and Methodology

GUVs were prepared using the electroformation method [15]. Pure POPC or 2 mol % opto-lipid (Opto-DArG/Pho-DAG-1) containing the POPC mixture were dissolved in chloroform to a final concentration of 4 mM. Then, 7 μL of these lipid stocks were spread on a pair of indium tin oxide (ITO)-coated glass slides (Delta Technologies Ltd., Loveland, USA) and dried in a desiccator for 2 h to evaporate the organic solvent. Afterwards, a 2 mm thick Teflon spacer was placed between two ITO glasses (with the conducting sides facing each other) and the chamber formed in this way was filled with 100 mM sucrose solution containing 0.5 mM NaCl. The ITO slides were connected to a function generator (Agilent, Waldbronn, Germany) to start the electroswelling process by applying an alternating current (AC) field (1.6 Vpp and 10 Hz) for 1 h. The obtained vesicles were collected into glass vials and used fresh within 24 h after preparation. Before the microscopy observations, the GUVs were 10-fold diluted with 105 mM glucose solution to release the initial vesicle tension, render them quasi-spherical, and allow for phase-contrast imaging. The vesicle suspension was then transferred into an electrofusion chamber (Eppendorf, Hamburg, Germany) and the vesicles were observed using an inverted AxioObserver D1 microscope (Zeiss, Munich, Germany) equipped with a Ph2 20×/0.4 objective and a pco.edge sCMOS camera (PCO AG, Kelheim, Germany). The GUVs were observed using a halogen lamp and a 546 nm filter introduced above the condenser to avoid any UV/blue light influence coming from the lamp, a state which we will refer to as “dark”. When necessary, the vesicles were illuminated with UV (365 nm) or blue (430 nm) light using respective filters and an HBO 100 W mercury lamp in epi-illumination.

In order to monitor the photoisomerization effects and light-induced morphological changes in photolipid-containing membranes, the GUVs were exposed to an electric field (5 Vpp and 1 MHz), which deforms them into prolate ellipsoids [16,17], pulling out the membrane area stored in fluctuations and allowing the detection of area changes associated with UV or blue light irradiation. The electric field was necessary to prevent uncontrollable vesicle shape changes, such as distortion of quasi-spherical (oblate) or elliptical structures, budding, tubulation, etc., and to allow quantitative monitoring of the vesicle geometry and membrane area [18,19]. GUV contours were detected in the phase-contrast images using a home-developed program [20], and the a/b aspect ratio was assessed. Because the preparation protocol does not allow controlling the initial vesicle size and excess area, the response to the electric field results in a different degree of deformation from vesicle to vesicle. Thus, we averaged the results from vesicles of a roughly similar size and similar changes in the aspect ratio (similar excess area) for specific membrane composition to allow comparison of the effect of light-induced changes.

### 2.5. Statistics

Data analysis and graphical display were performed using Clampfit 11.1 (Axon Instruments; Molecular Devices, San Jose, CA, USA) and SigmaPlot 14.0 (Systat Software Inc.; Chicago, USA). In order to measure the time constant of blue light-induced current deactivation, the technical delay (10 ms) of the system was subtracted. The data are presented as the mean values ± SEM. For normal distributed values (confirmed by Shapiro–Wilk test), Student’s two-sample *t*-test or the paired *t*-test was used to analyze the statistical significance. Equality of variances was tested using Levene’s test, and ANOVA was performed when appropriate. For non-normally distributed values, the Mann–Whitney rank-sum test was applied. In general, differences were considered significant at *p* < 0.05.

## 3. Results

### 3.1. Activation of TRPC Channels by Cis OptoDArG Promotes Thermal Relaxation of the Photochrome

A detailed analysis of the kinetic features of TRPC3 activation/deactivation cycling by photochromic diacylglycerols revealed a striking difference between two related DAG photolipids PhoDAG-1 and OptoDArG. Both molecules harbor azobenzene moieties within one (PhoDAG-1, Figure 2A) or both (OptoDArG, Figure 2D) of their fatty acid chains. UV light (365 nm) triggers *trans–cis* isomerization, and this transition is rapidly reversed by blue light (460 nm). Figure 2B,E illustrate the time courses of membrane conductance observed when the TRPC3 channels expressed in the HEK293 cells were activated by UV light (365 nm)-induced *cis* isomerization of photolipids followed by a period without illumination (dark) and subsequent blue light (430 nm)-induced *trans* isomerization. Both PhoDAG-1 and OptoDArG gave rise to rapid activation of the TRPC3 conductance upon UV illumination, as illustrated by the course of inward currents in Figure 2B,E. Upon termination of UV illumination, inward currents remained stable with PhoDAG-1 but not with OptoDArG as a lipid channel agonist. The decay of *cis* OptoDArG-induced currents in the dark was best described by a single exponential, similarly to the more rapid decline triggered by blue light, reflecting current deactivation. The second period of UV illumination restored the OptoDArG-induced conductance after its decline in the dark, while no additional effects of UV light were observed in the presence of PhoDAG-1. Of note, activation kinetics were typically faster for the second activation cycle. This phenomenon reflects a lipidation-dependent sensitization process that was described recently [21].

To test if the decline of *cis* OptoDArG-induced TRPC3 currents in the dark was based on thermal relaxation of the *cis* isomer within the plasma membrane, resulting in the dissociation of *cis* OptoDArG from its binding pocket in the channel complex, we investigated the thermal stability of *cis* photolipids in lipid bilayers. *Cis–trans* isomerization of photochromic DAGs in a lipid bilayer can be monitored by following the morphological response of cell-sized giant unilamellar vesicles (GUVs) [22] as a readout (see, e.g., [13,18]). We measured the degree of membrane deformation of GUVs containing 2 mol % photochromic DAGs (PhoDAG-1, Figure 2C; OptoDArG, Figure 2F). The vesicles were exposed to an electric field during an illumination protocol corresponding to that applied in electrophysiological experiments (Figure 2B,E). The main difference was that in the GUV experiments, transition from UV to blue light required a manual change of filters (Figure 2C,F), which resulted in an additional 10 s window between the termination of UV and onset of blue light illumination, however, with no apparent impact on the final result.

First, an electric (AC) field was applied 10 s prior to starting the irradiation protocol. This step ensures that the hidden area stored in fluctuations, membrane nanotubes, or defects, is pulled out while rendering vesicles with a well-defined prolate ellipsoidal shape, which facilitates the analysis of such experiments (see Appendix A for example vesicle snapshots and definition of the a/b aspect ratio). The GUVs were then irradiated with UV light at 10 and 50 s and with blue light at 70 s from the beginning of an experiment. However, due to the manual character of the experimental setup and averaging of several individual experiments, a shift of ± 2 s is visible in the graphical representation of the onset of light irradiation and the corresponding response. The experiments lasted 140 s (2 min 20 s) in total.

For a simpler comparison of the patch clamp (live cell) and GUV measurements, Figure 2C,F show excerpts of the experiments, displaying only the effect of UV/blue light irradiation whilst disregarding the effect of the electric field (the corresponding plots also featuring the effect of the electric field are shown in Appendix A). In both cases, the graphs represent an average of the traces of nine different vesicles. The GUVs containing either photolipid displayed rapid morphological changes upon UV and blue light illumination corresponding to the observed channel activation and deactivation recorded in the HEK293 cells, respectively.

However, Figure 2C,F demonstrate that none of the photolipids exhibited a significant relaxation in terms of reversal of its impact on the lipid bilayer structure after termination of UV light, indicating that the *cis* isomer remained fairly stable in the dark. This obvious stability of *cis* OptoDArG in the lipid bilayer led us to hypothesize that thermal relaxation of the photochrome is promoted by its interaction with the target (TRPC3) protein structure. Such a mechanism of facilitated thermal relaxation of *cis* OptoDArG by the protein environment is anticipated to display sensitivity to alterations in the direct binding environment of the DAG within the channel complex.

Interestingly, we detected a slight degree of vesicle deformation when comparing in the absence of irradiation at the beginning and in the final stage of the experiment (as seen in Figure 2C,F). In order to verify that the effect does not emerge from UV/blue light irradiation due to chemical modifications of photolipids (e.g., oxidation), we performed control experiments with pure POPC GUVs. A similar effect was observed in both the control experiments that featured UV/blue light irradiation and the ones that were performed only in the presence of the AC field (no UV/blue light), as shown in Appendix A. Presumably, out-of-focus structures or submicroscopic defects, such as tubes, give rise to the deformation as they are slowly pulled out to the vesicle surface over time due to the influence of an AC field.

### 3.2. Kinetics of Thermal Relaxation of OptoDArG within TRPC Complexes Provides Information on Lipid-Binding Structures and Molecular Mechanisms of Channel Regulation

In the next step, we therefore set out to compare the kinetics of the conductance decay seen in the dark with OptoDArG as an agonist for three closely related DAG-gated TRPC channels (TRPC3, TRPC6, and TRPC7). We hypothesized that the structural differences in the DAG interaction domain designated as L2 [2,23,24], which was recently identified for TRPC3 and TRPC6, will give rise to detectable changes in thermal relaxation. Figure 3 shows a comparison of the time courses of current deactivation in blue light (Figure 3A–C) and its relaxation in the dark (Figure 3D–F) for different TRPC channel isoforms (TRPC3/6/7). We found a significant difference between the channel isoforms regarding the time constants of the monoexponential current decay for both the fast blue light-induced current deactivation and even larger differences for the slow current decay in the dark (Figure 3E,F). Characteristic times describing the slow decay were significantly different between TRPC3 and TRPC6 as well as between TRPC6 and TRPC7 (Figure 3F). These results were consistent with the hypothesis of lipid–protein interaction-dependent thermal relaxation of the photochrome when bound to the channel, a phenomenon that is anticipated to report on differences in DAG–protein interactions.

Next, we challenged this concept by testing if local changes at the OptoDArG-binding site L2 introduced by point mutations would be detectable by altered kinetics of OptoDArG *cis–trans* relaxation in the dark. We generated two mutations in the L2 lipid-binding region, which were previously found to interfere with DAG regulation and proposed as the key residues for DAG coordination [21].

Figure 4 illustrates the consequences of single point mutations within the L2 region of TRPC3 on the kinetics of the decay of OptoDArG-mediated inward currents in the dark. Both mutations (position of the mutated residues in L2 is illustrated in Appendix A), which were previously shown to affect lipid gating [2,21], generated clear changes in the relaxation process by promoting the efficiency of channel deactivation. We measured the changes in the time constant and/or the maximum level of relaxation in the dark. Introducing the G652A mutation into the pore domain of the channels, which reportedly exerts a substantial change in the regulation of the channel by endogenous DAG and OptoDArG [2,21], significantly affected both the characteristic time (Figure 4C) and the level of recovery towards the basal conductance (Figure 4D). The Y648F mutant notably altered the level of recovery (Figure 4D) without significantly affecting the characteristic time of recovery. As illustrated in Appendix A, four point mutations within L1, representing an alternative lipid coordination site within TRPC3, failed to affect the characteristic time of the monoexponential decay of OptoDArG-activated TRPC3 in the dark.

## 4. Discussion

Our results suggest that quantifying the thermal relaxation process of photolipids is a valuable approach to gain information on the molecular basis of lipid–protein interactions. We showed that photochromic diacylglycerol OptoDArG displays a profound thermal relaxation of the active *cis* conformation to the inactive *trans* conformation. This process was found to be dependent on the structure of the DAG-binding domain in TRPC channels. Thermal relaxation of OptoDArG was about two orders of magnitude slower than blue light-induced *cis–trans* isomerization and also less complete. Incomplete current decay to an enhanced level of basal activity may reflect a state of partial lipidation of the tetrameric channel complex by *cis* OptoDArG as recently proposed [21]. Of note, both blue light-induced channel deactivation and thermal relaxation were best fit by single exponentials—consistent with the current model for gating of tetrameric TRPC channels by DAG involving conformational transitions within each subunit of the complex [2]. 

Our results support the notion that both blue light-induced *cis–trans* isomerization and thermal *cis–trans* relaxation of OptoDArG occur within the channel complex. Consistently, we observed a similar pattern of isoform dependence for kinetics of blue light-induced deactivation and thermal relaxation. This may be interpreted in favor of the concept that both processes are governed by the lipid-binding structure of the channel. The slow exponential decay of TRPC currents upon termination of UV illumination was observed with OptoDArG but not if using PhoDAG-1 as the agonist. This may be explained by the structural difference of these photolipids. OptoDArG undergoes a more extreme structural change in photocycling because harboring azobenzene moieties within both acyl chains is likely to reduce stability of the *cis* isomer within the TRPC’s lipid binding pocket. The hypothesis of target protein dependence of facilitation of thermal relaxation was corroborated by a comparison of different DAG-activated TRPC isoforms and by mutagenesis of the putative DAG interaction site. It is important to note that the architecture of the proposed DAG-binding region L2 [23] is well-conserved in TRPC channels, and this lipid coordination site is strikingly similar in closed TRPC3, 6, and 7 channels. Nonetheless, as the deactivating isomerization process is presumably initiated in the active/open conformation, the observed sensitivity of the thermal relaxation of bound *cis* OptoDArG may be interpreted in terms of more accentuated isoform differences within L2 of the channel’s open state. Mutagenesis in the recently discovered DAG interaction site L2 [23] significantly affected thermal relaxation of OptoDArG. This phenomenon was not observed for three mutations in an alternative lipid interaction domain (L1) of TRPC3, and only one mutation displayed a slightly altered level of current recovery by thermal relaxation. Hence, analysis of the kinetic features of thermal relaxation is potentially valuable to probe lipid–protein interactions. Using this approach, we confirmed here the current concept of DAG interaction with the L2 domain of TRPC channels [2]. Moreover, our present findings open the view on achieving unique specificity of photopharmacological interventions by taking advantage of the divergent thermal relaxation of certain *cis* photolipids when bound to different target proteins while being rather stable in the lipid bilayer. We indeed present evidence for *cis* OptoDArG being fairly stable when residing in a protein-free lipid bilayer over the typical recording times in our experiments. Since thermal relaxation in the biomembrane is essentially slow, deactivation of TRPC channels by dissociation of *cis* OptoDArG is unlikely to contribute to the observed current decline in the dark.

In summary, OptoDArG-activated TRPC channels or mutants show divergent kinetics of current decay in the dark and links to a specific impact of the DAG-binding site structure on thermal relaxation. Consequently, the use of distinct temporal photocycling protocols comprised of UV light pulses with optimized duration and frequency is likely to enable TRPC isoform-selective modulation of endogenous TRPC conductance. Similarly, this concept may allow controlling other lipid-regulated cellular signaling pathways with light in a highly specific manner.

## 5. Conclusions

We suggest that the analysis of the functional changes associated with thermal relaxation of OptoDArG can provide information on the involved DAG coordination site of the target signaling proteins. The use of photochromic lipids with distinct thermal relaxation features in combination with optimized temporal illumination pattern is proposed to enable highly specific modulation of cellular signaling pathways.

## Figures and Tables

**Figure 1 biomolecules-12-00799-f001:**
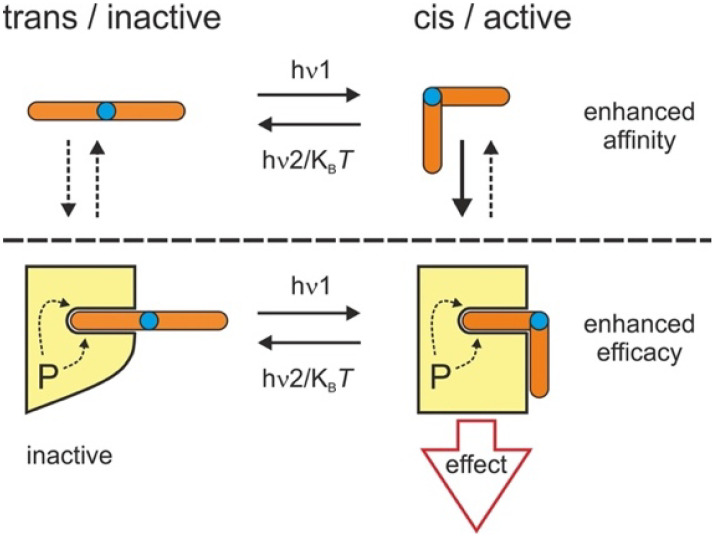
Hypothetical scheme of how azobenzene photolipids control their target proteins via a combined affinity and efficacy switch mechanism. Photoisomerization of photolipids between *cis* and *trans* potentially modifies both affinity and efficacy of photoligand action. As found for photochromic DAGs [2,3], the scheme depicts a scenario in which a light-induced (h√1) *cis* isoform exerts higher affinity as well as higher efficacy on its target, thereby initiating a productive change in this molecule (effect). That is why the “off-kinetics” in terms of returning the target to its inactive state may involve either dissociation of the *cis* isoform via its removal from the target´s surroundings as a result of *cis–trans* isomerization, either light-induced (h√2) or by thermal relaxation (K_B_T), or, alternatively, due to the relaxation of the *cis* isoform (h√2/K_B_T) within its binding pocket.

**Figure 2 biomolecules-12-00799-f002:**
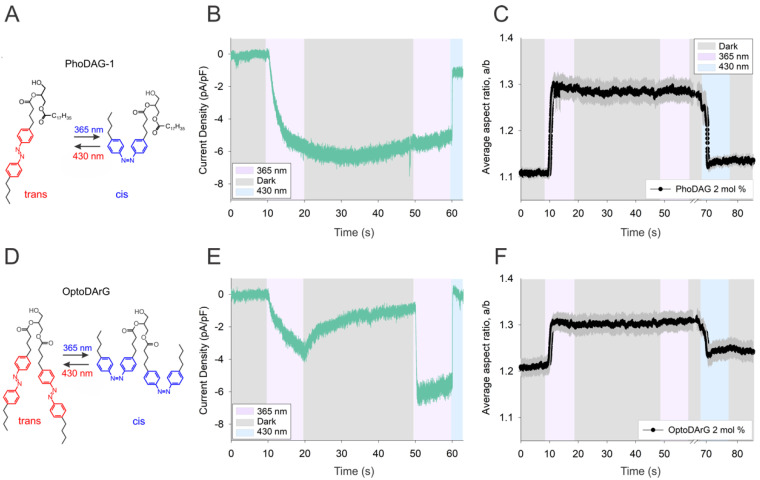
Comparison of the kinetics of thermal relaxation of photochromic diacylglycerols. (**A**,**D**) Chemical structures of photochromic diacylglycerols: (**A**) PhoDAG−1; (**D**) OptoDArG. (**B**,**E**) Representative traces showing the inward currents induced by photoactivation of PhoDAG−1 ((**B**), 100 μM) and OptoDArG ((**E**), 20 μM) in a whole−cell gap−free recording (holding potential: −40 mV, baseline adjusted to 0 mV, normalized by capacitance) in TRPC3−WT−expressing HEK293 cells. To photoconvert OptoDArG, we used 100% intensity of UV (365 nm; violet) for 10 s to switch to active (*cis* state), followed by the dark (UV illumination off) for 30 s (grey) and then blue light (430 nm) for 3 s (blue). (**C**,**F**) GUV shape deformation (presented in terms of the a/b aspect ratio) due to photoisomerization of (**C**) PhoDAG−1 and (**F**) OptoDArG, each averaged over nine different vesicles (the standard error of mean is shown in dark gray). The same irradiation protocol as in (**B**,**F**) was followed.

**Figure 3 biomolecules-12-00799-f003:**
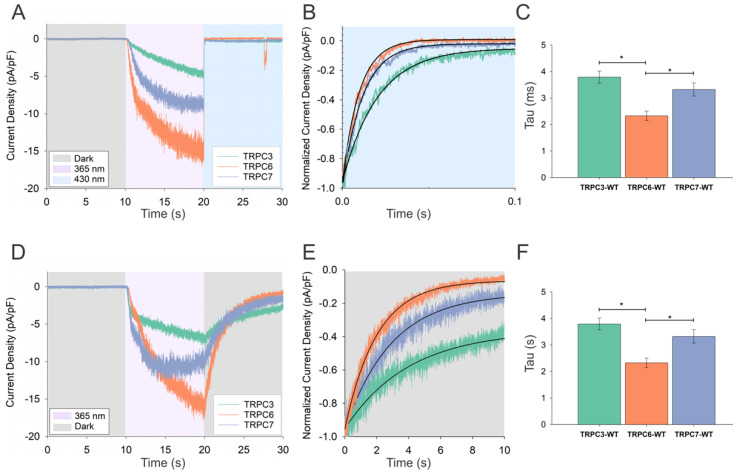
OptoDArG−activated TRPC isoforms (TRPC3–TRPC6–TRPC7) display variable decay kinetics in blue light and in the dark. (**A**) Representative recording showing the gating kinetics induced by instantaneous photoisomerization of OptoDArG (20 μM) to *cis* using UV light (365 nm, 10 s; violet) and then to *trans* conformation using blue light (430 nm, 10 s; blue) in a whole−cell gap−free recording (holding potential: −40 mV, baseline adjusted to 0 mV, normalized by capacitance). (**B**) Exponential fit of the representative normalized time courses of TRPC3 (green), TRPC6 (orange), and TRPC7 (blue) current deactivation during blue light illumination. (**D**) Representative current traces showing gating kinetics induced by thermal relaxation of OptoDArG (30 µM) in TRPC3 (green), TRPC6 (orange), and TRPC7 (blue) after UV illumination was switched off (dark). (**E**) Exponential fit of representative normalized time courses of TRPC3 (green), TRPC6, (orange) and TRPC7 (blue) current deactivation after UV illumination was switched off. (**C**,**F**) Bar charts illustrating the deactivation time constant (Tau) for blue light (**C**) and after UV was switched off (dark) (**F**) for TRPC isoforms. Number of biological repetitions for each condition ≥ 6. The data are the means ± SEM; two-tailed *t*−test or Mann–Whitney test was applied; * *p* < 0.05.

**Figure 4 biomolecules-12-00799-f004:**
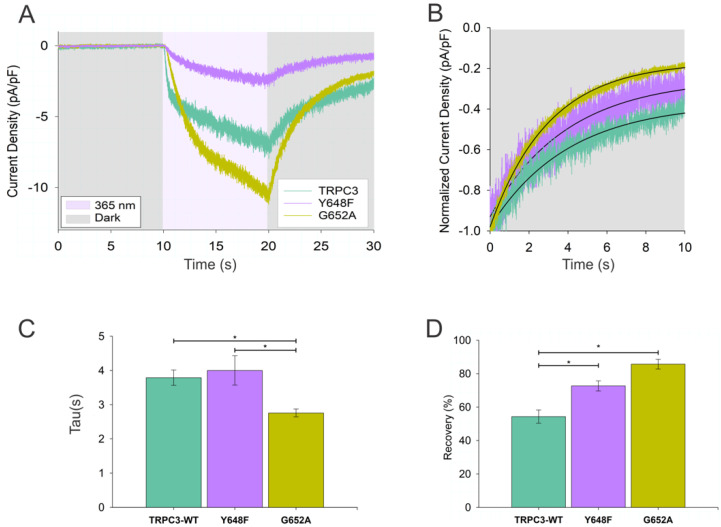
Characteristics of TRPC3 channel thermal relaxation-mediated deactivation is altered by mutation in the L2 region. (**A**) Representative current traces showing gating kinetics induced by thermal relaxation of OptoDArG (30 µM) in TRPC3−WT (green), Y648F (violet), and G652A (yellow) after UV illumination was switched off (dark) in a whole−cell gap-free recording (holding potential: −40 mV, baseline adjusted to 0 mV, normalized by capacitance). (**B**) Exponential fit of representative normalized time courses of TRPC3-WT (green), Y648F (violet), and G652A (yellow) current deactivation after UV illumination was switched off (dark). (**C**,**D**) Bar charts illustrating the deactivation time constant (Tau; (**C**)) and the percentage of the recovery to basal conductance after UV was switched off (dark; (**D**)); TRPC3−WT and TRPC3 mutants. Number of biological repetitions for each condition ≥ 6. The data are the means ± SEM; two−tailed *t*−test or Mann–Whitney test was applied; * *p* < 0.05.

## Data Availability

This study includes no data deposited in external repositories. The original data related to this paper will be provided upon reasonable request by the corresponding authors.

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
