# Peer review of "Characterization of DAG Binding to TRPC Channels by Target-Dependent cis–trans Isomerization of OptoDArG"

_biomolecules, 2022, doi:10.3390/biom12060799_

Round 1
Reviewer 1 Report
Light-induced isomerization of synthetic photochromes such as azobenzene-based photochromic lipids are useful probes for the analysis of ion channel-lipid interactions. Rapid photoisomerization of these molecules enables the analysis of lipid gating kinetics and provides information on lipid-sensing. Using visualized morphological response of giant unilamellar vesicles giant liposome, Groschner and colleagues exploited cis-photolipid-induced changes in pure lipid membranes to show that thermal cis-trans isomerization of both PhoDAG-1 and OptoDArG is essentially slow in the lipid bilayer environment. In the present study they exploited azobenzene-based photochromic lipids, biologic activity linked to the cis conformation, which, for standard azobenzenes is thermodynamically less stable and returns to the trans state by thermal relaxation with variable kinetics. Rapid trans-cis photoisomerization of azo-diacylglycerols (DAGs) enabled fast alterations in the active ligand concentration at the target protein. This technique was found suitable to delineate the kinetics of DAG activation of TRPC channels, and to explore the structural basis of channel regulation by DAG, as well as mechanistic aspects of channel regulation by the lipid. The authors found that the photochromic diacylglycerol OptoDArG displays prominent cis-trans relaxation when bound to TRPC channels. The kinetic of the thermal relaxation of this photochromic channel activator was found dependent on the channel isoform and the amino acid sequence of the predicted lipid binding domain. They proposed analysis of the kinetics of photolipid-mediated modification of channel functions as a strategy to identify structures and principles underlying their lipid regulation. Moreover, they reported TRPC isoform-dependent thermal relaxation of the channel activator cis-OptoDArG may serve as the basis of novel protocols for selective photopharmacological control over TRPC conductance's.
Comments
This is an important and well-done study and I only have few minor questions on the manuscript.
- In Fig. 2, how do you explain the difference between the current responses to the two consecutive UV lights in the OptoDArG experiment?
- Can you explain the differences in decay kinetics in blue light and in the dark among the 3 TRPC isoforms (Fig. 3) based on their amino acid sequence? What do you mean by the statement " the observed sensitivity of the thermal relaxation of bound cis-OptoDArG, may be interpreted in terms of more accentuated isoform differences within L2 of the channel´s open conformation."
- Did the G652L mutation completely eliminate the TRPC3 current?
- Can you compare the current sensitivity to application of DAG (SAG or OAG) and OptoDArG?
Reviewer 2 Report
The present manuscript investigates DAG-TRPC channel interaction using cis-trans photoisomerization of PhoDAG-1 and OptoDArG. The authors show that currents activated by cis-PhoDAG remained stable upon termination of UV light exposure while cis-OptoDArG-induced TRPC3/6/7 activity displayed an isoform-dependent exponential decay. The authors compared the kinetics of the conductance decay in the dark with OptoDArG as an agonist for TRPC3, TRPC6 and TRPC7 and report that point mutations within the L2 lipid coordination region altered the kinetics while four points mutations in L1 had no effect. The issue investigated is interesting and provides valuable information about DAG-TRPC channel interaction. The manuscript is well written and the study is carefully performed. There are some minor issues that need to be addressed.
1. In Fig 2E, the second exposure to 365 nm enhances TRPC3-mediated current, could the authors explain this phenomenon? Is it mediated by sensitization of the channel?
2. Figures should be embedded in the correct place within the text.
3. p. 353 “was found to be dependent”
